# Sculpting Windows onto AuAg Hollow Cubic Nanocrystals

**DOI:** 10.3390/nano13182590

**Published:** 2023-09-19

**Authors:** Javier Patarroyo, Neus G. Bastús, Victor Puntes

**Affiliations:** 1Institut Català de Nanociència i Nanotecnologia (ICN2), CSIC, The Barcelona Institute of Science and Technology (BIST), Campus UAB, Bellaterra, 08193 Barcelona, Spain; javier.patarroyo@icn2.cat; 2CIBER en Bioingeniería, Biomateriales y Nanomedicina, CIBER-BBN, 28029 Madrid, Spain; 3Vall d’Hebron Institut de Recerca (VHIR), 08035 Barcelona, Spain; 4Institució Catalana de Recerca i Estudis Avançats (ICREA), 08010 Barcelona, Spain

**Keywords:** hollow nanocrystals, galvanic replacement reaction, surfactant effect

## Abstract

Using surfactants in the galvanic replacement reaction (GRR) offers a versatile approach to modulating hollow metal nanocrystal (NC) morphology and composition. Among the various surfactants available, quaternary ammonium cationic surfactants are commonly utilised. However, understanding how they precisely influence morphological features, such as the size and void distribution, is still limited. In this study, we aim to uncover how adding different surfactants—CTAB, CTAC, CTApTS, and PVP—can fine-tune the morphological characteristics of AuAg hollow NCs synthesised via GRR at room temperature. Our findings reveal that the halide counterion in the surfactant significantly controls void formation within the hollow structure. When halogenated surfactants, such as CTAB or CTAC, are employed, multichambered opened nanoboxes are formed. In contrast, with non-halogenated CTApTS, single-walled closed nanoboxes with irregularly thick walls form. Furthermore, when PVP, a polymer surfactant, is utilised, changes in concentration lead to the production of well-defined single-walled closed nanoboxes. These observations highlight the role of surfactants in tailoring the morphology of hollow NCs synthesised through GRR.

## 1. Introduction

Embracing the forms of porous nanocages and nanoframes, bimetallic AuAg hollow nanocrystals (NCs) are garnering significant interest across diverse domains, including nanocatalysis [1,2,3,4,5], biomedical imaging [6,7], controlled drug delivery [8], chemical sensing [9], and photothermal applications [10]. In these systems, the essence of their applicability lies in the precision with which the engineering of the hollow systems is achieved, as it fundamentally determines their optical, catalytic, and structural properties. One of the most employed techniques for producing them is the galvanic replacement reaction (GRR).

GRR involves the spontaneous transmetalation redox reaction between two different metal ions driven by their difference in reduction potentials. A more reactive metal cation in solution (Au) takes electrons from a less reactive metal in a sacrificial template (Ag), leading to its oxidation and dissolution. In contrast, the noble metal is reduced and deposited onto the surface of the template [11]. In the process, as time advances or as the Au/Ag ratio increases, single or multiple voids open at the NC surface, which serve as channels through which the template’s core is removed, forming hollow nanostructures. In the initial stages of the process, as the more noble metal is deposited and covers the template, the chemical potential between the deposited and the corroded metals increases. This leads to the formation of a pinhole at the interface, allowing a flux of cations to escape and preventing its closure until the core of the template is voided. The number and position of these pinholes determine the final structure of multichambered hollow NCs [12]. Typically, the pinhole closes once the corrosion of the template is complete, mainly when the reaction is conducted at high temperatures [13,14]. Once the hollow NC is formed, for high Au/Ag ratios, further corrosion triggers a de-alloying process, forming pores in the hollow NC surface. As more vacancies collapse, these pores gradually develop, small and polydisperse in size and shape, distributed across the surface. While the shape of the resulting hollow NCs is primarily determined by the starting template, wall thickness, composition, and porosity are controlled by the interfacial alloying and de-alloying processes associated with the GRR [15,16,17].

Despite the effort, it remains challenging today to precisely control the position of the void within the hollow structure, especially its size and location [18]. In the pursuit of optimising performance [19], significant efforts have been focused on creating a variety of hollow NCs with increasingly sophisticated surfaces and inner structures via the coupling GRR with sequentially deposited templates [14,20,21], the Kirkendall effect [12,22] and combining co-reduction and co-corrosion processes [23,24]. Following these strategies, different configurations have led to a plethora of hollow NCs. However, one challenge remains the rational control of the pore locations within these hollow structures. Often, pores are small and randomly distributed across surfaces [8,25]. Attempts to directly enlarge the pores often result in nanoframes forming with huge voids, risking structural collapse [14]. In this context, it has been observed that localising the pores at the more reactive sites, such as the corners of the cube-shaped NCs, allows for enlarging the pores without compromising the overall structural integrity of the NCs. Xia and co-workers, in 2006, motivated the use of Ag nanocubes with truncated corners to obtain AuAg nanocages with pores confined to the corners [26]. Later on, Sun et al. selectively deposited Ag_2_O patches at the corners of Ag nanocubes to chemically control the removal of Ag_2_O deposits at the corner sides and the Ag core, forming Au-based nanoboxes with well-defined openings at corners [27]. Lu et al. recently developed a one-step process to produce hollow NCs with well-defined openings at the corners by employing cetylpyridinium chloride [28]. The method is based on the well-known fact that molecular surfactants, when absorbed, can modify the surface energy of the NC, thereby influencing—favouring or hindering—the deposition of atoms on specific crystalline facets, ultimately affecting the growth kinetics and morphology of the NC [29]. Moreover, surfactants interact with the precursors and intermediate species, governing whole NC growth kinetics [30]. This has been extensively explored and exploited for anisotropic crystal growth since Peng and Alivisatos’ seminal work on CdSe nanorods [31]. However, understanding surfactant effects in the formation and evolution of NCs during reverse growth reactions, i.e., by removing ions from a template rather than adding them, as in GRRs, remains poorly described and understood.

Built upon the previous work of Gonzalez and co-authors [12] on the production of complex hollow NCs at room temperature, we herein study the role of the surfactant in shaping and controlling the morphological characteristics of hollow AuAg NCs, particularly the pore distributions at its surface. To this end, several surfactants, including three quaternary ammonium salt cationic surfactants, cetyltrimethylammonium bromide (CTAB), chloride (CTAC), and p-toluenesufonate (CTApTS), along with Polyvinylpyrrolidone (PVP), were tested. As a typical cationic surfactant, CTAB has turned out to be very effective in the shape-controlled synthesis of gold nanorods [32,33] and for the synthesis of complex hollow NCs [12]. CTAB and CTAC present the same alkyl chain but a different halide counterion. A non-halogenated cationic surfactant CTApTS was used to study the effect of the counterion on the final morphology. Additionally, a non-cationic polymer such as PVP was used to evaluate the role of the ionic surfactant.

Understanding the influence of diverse molecular components provides valuable information for the rational design and synthesis of complex hollow NCs with tailored structures and functionalities by adding different surfactant amounts. Results show that the halide counterion controls the void formation in the hollow structure. When halogenated surfactants, such as CTAB or CTAC, are employed, multichambered opened nanoboxes are formed. The main difference is that, when using CTAB, a pinhole forms at the centre of the nanocube face, while, when utilising CTAC, the pinhole emerges non-centred, away from the face’s centre, edges, and corners. On the other hand, when CTApTS was used, only single-walled closed nanoboxes with an irregular thick wall were obtained. Finally, when PVP was used, well-defined single-walled closed nanoboxes were obtained.

## 2. Results

The production of AuAg hollow nanocrystals (NCs) is a two-step process which involves the synthesis of monodisperse Ag nanocubes, which are later used as sacrificial templates for preparing hollow AuAg nanoboxes by GRR at room temperature in the presence of the different surfactants [5,16,22,34] (Appendix A). Ag NCs with well-defined shapes, such as nanospheres, nanocubes, nanorods, nanowires, and nanoprisms, are commonly used as sacrificial templates. Among them, we used cubic Ag NCs produced by polyol reduction as sacrificial templates due to their anisotropic shape, interesting optical properties, and widely described applicability [17].

Representative transmission electron microscopy (TEM) images of the as-obtained samples (Figure 1A) show the systematic formation of highly monodisperse Ag nanocubes, whose crystals exhibit an edge length of 45.4 ± 5.1 nm in the sides (Figure 1B), which falls within the typical range of sizes employed in GRR studies. The UV–Vis absorption spectra of colloidal Ag nanocubes display distinct and intense surface plasmon resonance (SPR) peaks at ~440 nm and ~350 nm, indicative of their narrow size distribution cubic shape [35,36,37] (Figure 1C). The X-ray diffraction (XRD) patterns reveal four distinct diffraction peaks at 2θ values of 38, 44, 64, and 76°, which can be attributed to the (111), (200), (220), and (311) fcc Ag crystal planes, respectively (JCPDS file No. 04-0783). The overdominance of the {200} reflection in the XRD pattern confirms the flat faces of the NCs and their cubic shape (Figure 1D) [38].

These synthesised Ag nanocubes were utilised as sacrificial templates for producing AuAg hollow nanoboxes. In a typical experimental procedure, the Ag nanocubes were dispersed in a reaction medium containing various surfactants, ascorbic acid (AA) employed as a mild reducing agent, and an aqueous HAuCl_4_ solution, which was slowly added into the reaction mixture using a syringe pump. The presence of AA in the reaction system enables the reduction of AuCl_4_^−^ into AuCl_2_^−^. This co-reduction process competes with the transmetalation redox process. It modifies the redox potential of Au cations, reducing their etching power and, ultimately, allowing for tailoring the structural features of the resulting NCs at both its morphology and roughness and porosity of the walls. The reaction temperature plays a crucial role in forming bimetallic AuAg nanostructures. The GRR rate is faster at higher temperatures, often leading to more aggressive corrosion processes, affecting the final structure’s roughness and porosity [39], pinhole sealing [12], and loss of reproducibility. Thus, to have better control over the GRR’s kinetics and the products’ morphology, all our experiments were performed at room temperature (RT).

The structural evolution of the Ag nanocubes in the presence of CTAB (14 mM) with the addition of different amounts of HAuCl_4_ (1 mM) was investigated. CTAB is a cationic surfactant widely used in the shape-controlled synthesis of noble metal NCs. It acts as a stabiliser, structure-directing agent, and complexing ligand, effectively reducing the reduction rate of the metal precursor while facilitating its epitaxial deposition [33]. During the reaction, the morphological and compositional changes of Ag nanocubes were studied by high-angle annular dark-field scanning transmission electron microscopy (HAADF-STEM), energy-dispersive X-ray spectroscopy analysis (EDS), and UV–Vis spectroscopy. When substoichiometric amounts of HAuCl_4_ (25 μL, 6.25 × 10^−4^ mmol) were added to the Ag nanocube solution, a thin layer of Au was initially deposited on their surfaces (Figure 2A). This deposition occurred through the direct reduction of Au cations to Au^0^ in solution, facilitated by the action of AA [40]. As more Au precursor was added (50 μL, 1.25 × 10^−3^ mmol), a small pinhole formed at the centre of the faces of the nanocubes (Figure 2B), indicating that the reaction is initiated locally at the high-energy sites of the crystal surface rather than randomly across its entire surface. As the reaction proceeded (75 μL, 1.87 × 10^−3^ mmol), this small hole provided access to the Ag core, the anode, where Ag is oxidised, and electrons stripped and transported from the core through the solid to the NC surface where Au cations are reduced (Figure 2C). With the addition of Au (100 μL, 2.5 × 10^−3^ mmol), the conversion of the Ag solid nanocubes into hollow AuAg nanoboxes is achieved (Figure 2D). When higher amounts of HAuCl_4_ solution were used (150 μL, 3.2 × 10^−3^ mmol), double-walled opened structures were formed (Figure 2E–H), which exhibited the homogeneous distribution of Ag (31.3 ± 7.9%, red) and Au (68.7 ± 7.8% Au, green) (Figure 2I). As expected, its size (51.0 ± 4.7 nm) is more significant than that of the precursor Ag NCs (45.4 ± 5.1 nm), accounting for the deposition of the Au layer onto the sacrificial template structure.

One of the most exciting features of these bimetallic AuAg hollow nanostructures is the strong dependence of the colour of the colloidal solution on the degree of voiding of the NCs, which can be directly controlled by the amount of HAuCl_4_ added to the reaction. Thus, by increasing the content of HAuCl_4_, the solution gradually turns from yellow, dark yellow, orange, red, red-violet, purple, and, finally, blue (Figure 2F)**.** This colour evolution directly correlates with the position of the SPR band, allowing the precise tuning of the SPR peak to virtually any position within the visible range by controlling the volume of the added HAuCl_4_ (Figure 2G). Accordingly, while Ag nanocubes exhibit a narrow SPR band peaking at 440 nm, the double-walled opened nanoboxes display a broader band centred at ~680 nm. While the redshift is directly attributed to the formation of hollow structures with increasing Au content [41], the broadness of the peak can be ascribed to both phase retardation effects in larger NCs at lower energies [42]. The purity of the final structures, where single- and double-walled AuAg nanoboxes with different edge lengths, thickness, and corner sharpness coexist in the sample, can be observed in Figure 2E. This remarkable tunability of both colour and SPR peak position in the visible region highlights the unique optical properties of the bimetallic multichambered opened AuAg nanoboxes, providing opportunities for diverse applications in fields such as sensing, imaging, and plasmon-mediated chemical reactions.

In our system, the critical role of CTAB assisting the controlled transmetallation process is revealed when its concentration was systematically varied by three orders of magnitude, from 0.14 mM to 140 mM, while keeping constant the amount of HAuCl_4_ and Ag templates added to the reaction mixture. When the concentration of CTAB was kept low (0.14–1.4 mM), the process was less controlled, leading to poorly defined nanoboxes with compromised structural integrity and agglomerated small Au nanoparticles. In contrast, increasing the CTAB concentration to 140 mM facilitated a more controlled and efficient process, forming well-defined nanocrystals with single face-centred holes. (Appendix A).

The halide ions also play essential roles in directing the anisotropic growth of solid metal NCs. To investigate their effect on the final morphology of hollow NCs, we substituted CTAB with CTAC at the same concentration (14 mM) while keeping all other experimental parameters unchanged. Halide ions, such as Cl and Br, not only modulate the redox potentials of metal ions but also form insoluble compounds with Ag, exhibiting a strong affinity for the {111} crystal planes rather than the {001} planes. This preference leads to the effective passivation of the corners and edges of the Ag nanocubes by the formation of AgBr compounds. The morphological evolution of the Ag nanocubes with increasing amounts of HAuCl_4_ added to the reaction media is shown in Figure 3A–D. The use of CTAC resulted in notable morphological differences, especially in the initial stages of the reaction (50 μL, 1.25 × 10^−3^ mmol). Thus, the pinhole appears non-centred, away from the nanocube’s centre, edges, and corners (Figure 3A–E). EDS maps of a single non-centred pinhole NC show that Au is homogeneously distributed throughout the crystal, forming an outer layer of approximately ~4 nm (Figure 3F).

In this case, by using higher amounts of HAuCl_4_ solution (125 μL, 3.2 × 10^−3^ mmol), we also observed the formation of double-walled opened nanoboxes (Figure 3D). Importantly, the non-centred position of the pinhole is maintained. When multiple pinholes are formed (Appendix A), multichambered opened nanoboxes are produced (Figure 3G). The average length of the resulting nanoboxes was 54.2 ± 4.3 nm, with a wall thickness of 7.0 ± 1.4 nm (Appendix A), almost the same size as the nanoboxes obtained with CTAB. Remarkably, the inner cage is not centred, as observed with CTAB. Instead, its position corresponds to the pinhole’s initial location, which impacts the absorbance profile of the NCs (Figure 3I) [43]. EDS maps of a double-walled nanobox reveal a homogeneous distribution of Ag and Au throughout the crystal (Figure 3H). This observation is consistent with line-scanning profiles, where the nanoboxes consist of both Ag and Au, with Ag being more abundant than Au. Furthermore, both hollow cavities are distinctly visible in the HAADF-STEM image and the EDS profile, where a drop in the Ag signal confirms their presence (Figure 3J,K).

We hypothesise that, in the presence of Cl-, the compact ordering of AgCl (note that Br^−^ is significantly larger than Cl^−^) loosens in density as it approaches corners and edges. As a result, far from the centre, the area better protected by the AgCl flat layer, and close but not at the corners or edges, for which higher reactivity leads to faster passivation, is at these distances where GRR better occurs. This ultimately leads to pinholes opening away from the centre of the crystal face and closer to the corners and edges. This contrasts with hollow NCs synthesised at higher temperatures [11] or using PVP as a stabiliser, where the pinhole tends to close. As previously mentioned, carrying out the reaction at low temperatures is desirable to preserve the open pinhole and avoid surface reconstruction. Notably, if the reaction is performed more aggressively (rapid addition of more HAuCl_4_), multiple pinholes may appear, consistently positioned away from the corners and the center. Four windows can be opened in each cube face in such cases, leading to square-cross 3D morphologies.

The optical properties of the synthesised AgAu nanoboxes were investigated using UV–Vis spectroscopy (Figure 3I). The SPR peak exhibited a redshift, ranging from ~440 nm to ~650 nm as the concentration of HAuCl_4_ solution was increased (1.25 × 10^−3^ mmol to 3.12 × 10^−3^ mmol). As mentioned earlier, this redshift results from the increasing Au content in the nanostructure and the enlargement of the void size. Similar to the observations with CTAB, the broadness of the SPR band is attributed to both phase retardation effects in Au-rich structures and the formation of various products during the reaction, including single-, double-, or multi-walled nanoboxes, as well as the presence of different morphologies.

Next, we investigated the effect of a non-halogenated cationic surfactant, specifically p-toluenesulfonate (CTApTS) (Figure 4). When CTApTS was used as the surfactant, single-walled closed nanoboxes with an average edge size of 53.1 ± 5.0 nm and irregular thick walls of 8.3 ± 1.7 nm were obtained (Appendix A). Notably, the characteristic pinhole previously observed with other surfactants containing halogenated counterions was not seen in this case. This suggests that galvanic corrosion occurs rapidly during the early stages of the reaction. Single-walled and partially corroded nanoboxes further support this observation. As more gold precursor was added, the formation of poorly controlled single-walled nanoboxes increased in abundance. The UV–Vis spectra evolution with the addition of different amounts of Au precursor is depicted in Figure 4E. Initially, upon adding small volumes of the Au precursor, the SPR band redshifts until approximately ~550 nm. As more gold precursor is added, the spectra transform into a flat line with a slight bump around ~750 nm. This noticeable change in the absorption spectra is attributed to the aggregation of the resulting nanostructures.

The obtained results indicate that the presence of halogens provided by the surfactant forms a layer of AgBr (Ksp: 5.4 × 10^−13^) or AgCl (Ksp: 1.8 × 10^−10^) on the surface of the Ag nanocubes with varying densities, which plays a crucial role in hindering fast corrosion and modifying the reaction kinetics. Specifically, AgBr or AgCl acts as a physical barrier, interfering with the galvanic reaction. Consequently, it exerts control over the diffusion of the Au precursor, favouring the formation of complex nanostructures [44].

Finally, the effect of a non-ionic stabiliser, PVP, was also investigated as it plays a pivotal role as a stabilising and shape-directing agent in synthesising noble metal NPs [45,46]. The influence of varying PVP concentrations was examined to ascertain the optimal precursor amount to induce the transmetallation reaction, akin to the earlier experiments with CTAB. PVP with a molecular weight of 55,000 was utilised at different concentrations while keeping the Au precursor concentration constant (100 μL of HAuCl_4_ 1 mM). Obtained results show that the concentration of PVP significantly influences the final nanostructure morphology, allowing the precise control of single-walled nanoboxes with high yield (Figure 5). Thus, when no PVP was employed (Figure 5A), the rapid oxidation of the Ag template, combined with the swift deposition of Au, resulted in the formation of sponge-like nanostructures. However, fixing the PVP concentration at 0.01 mM produced a mixture of sponge-like nanostructures and single-walled nanoboxes (pinholed or not) (Figure 5B). Notably, adjusting the PVP concentration to 0.1 and 1 mM (Figure 5C,D) yielded predominantly single-walled nanoboxes with high efficiency. As the PVP concentrations were further increased to 10 mM and 100 mM (Figure 5E,F), an inhomogeneous mixture of single-walled nanoboxes and partially corroded nanostructures was obtained. This phenomenon can be attributed to the PVP binding to the surface of Ag nanocubes, acting as a physical barrier that hinders Au precursor deposition–reduction, thereby slowing the kinetics of GRR [30]. HR-TEM images (Figure 5G) of single-walled nanoboxes shown in Figure 5D reveal the high monodispersity and homogeneity of the product. The EDS elemental mapping (Figure 5H) shows the distribution of the constituent elements in the sample, revealing that the wall of the nanobox consisted of 37.8 ± 6.2% Ag and 62.2 ± 6.2% Au, while the inner part consisted of 36.2 ± 9.2% Ag and 63.8 ± 9.2% Au. Remarkably, the walls and inner parts exhibited similar chemical compositions, enabling their presentation as an overall result: 37.8 ± 7.1% Ag and 62.2 ± 7.1% Au. The UV–Vis spectra of the colloidal solutions obtained at increasing concentrations of PVP are shown in Figure 5I. Unexpectedly, the structures obtained at higher PVP concentrations displayed a lower redshift (670 nm) than those obtained at lower PVP concentrations (up to 700 nm). This phenomenon can be attributed to the presence of nanostructures that are not fully corroded.

Interestingly, by employing even higher concentrations of PVP (180 mM) and a slow addition of the Au precursor (10 μL/min), we achieved the synthesis of highly porous nanoboxes (Figure 6A), which resulted from the de-alloying and dissolution of Ag. Selected TEM tilt images of the same nanobox (+50° to −45°) confirm the highly porous morphology of the nanoboxes (Figure 6B), further accentuating their unique structural properties. Remarkably, these nanoboxes exhibited a robust SPR band in the near-infrared region at 1000 nm (Figure 6C), rendering them particularly promising as contrast agents for photoacoustic tomography [47,48] and photothermal therapy [49].

## 3. Conclusions

We herein studied the effect of different surfactants on the synthesis of AuAg nanoboxes, showing how the halide counterion (Cl vs. Br) has an apparent impact in shaping the final morphology of hollow NCs, revealing the significant role of the halide counterion (Cl vs. Br) in controlling the formation of voids within the structure. The choice of surfactant and the amount of Au precursor added to the reaction were critical in shaping the nanostructures and determining the final morphology of the resulting nanoboxes and their optical properties. With CTAB, the SPR band exhibited a redshift from 440 nm (Ag nanocubes) to 680 nm (nanoboxes) as the amount of HAuCl_4_ increased. CTAC showed a similar process but at a faster rate, owing to the weaker interaction strength of AgCl compared to AgBr. On the other hand, using CTApTS resulted in the polydisperse etching of Ag templates and non-homogeneous deposition of Au, leading to optical absorption extending across the entire visible spectrum, with a slight bump around 750 nm. Employing high concentrations of PVP resulted in highly porous nanoboxes exhibiting a robust SPR band in the near-infrared region at 1000 nm.

Results show that the halide counterion controls the void formation in the hollow structure, which can be attributed to how halide ions modulate metal ion redox potentials and form insoluble compounds with Ag, selectively passivating the surface of Ag nanocubes. Thus, the presence of Cl^−^ induced a looser packing of surfactant molecules near corners, promoting reactivity and leading to pinholes opening away from the centre of the crystal face, ultimately forming multichambered structures. In contrast, using CTApTS as a surfactant produced single-walled closed nanoboxes with irregular thick walls. At the same time, PVP yielded well-defined single-walled closed nanoboxes, which evolved into highly porous structures by Ag de-alloying and dissolution.

Overall, these findings highlight the critical role of surfactant and gold precursor amounts in tailoring the properties of AuAg nanoboxes, offering a versatile approach to engineering these nanostructures for specific applications. This effect highlights the significant influence of halogen-containing surfactants in shaping the morphology and composition of the resulting nanocrystals, offering promising prospects for tailored nanomaterial design.

## Figures and Tables

**Figure 1 nanomaterials-13-02590-f001:**
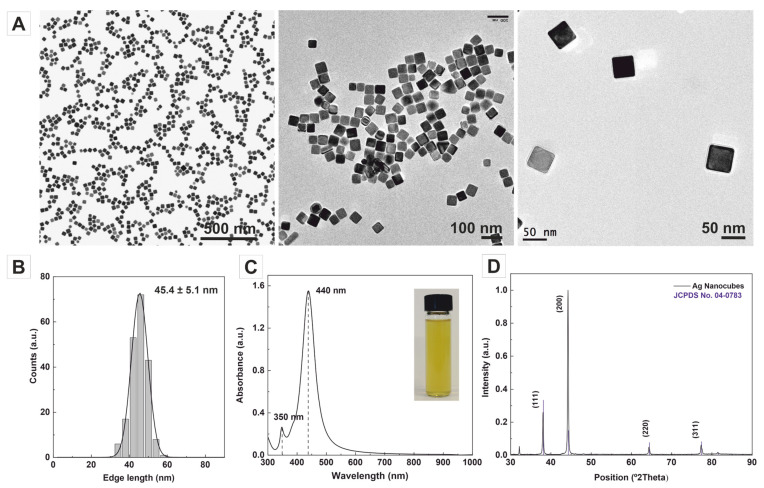
Synthesis of Ag nanocubes. (**A**,**B**) Representative transmission electron microscopy (TEM) images of the highly monodisperse Ag nanocubes with an edge length of 45.4 ± 5.1 nm. (**C**) UV–Vis absorption spectra exhibit strong SPR peaks at ~440 nm and ~350 nm, indicating the uniform size distribution of the nanocubes. The inset shows a colloidal sample of the AgNCs. (**D**) XRD patterns of Ag nanocubes with distinct diffraction peaks corresponding to specific crystallographic planes and JCPDS file No. 04-0783.

**Figure 2 nanomaterials-13-02590-f002:**
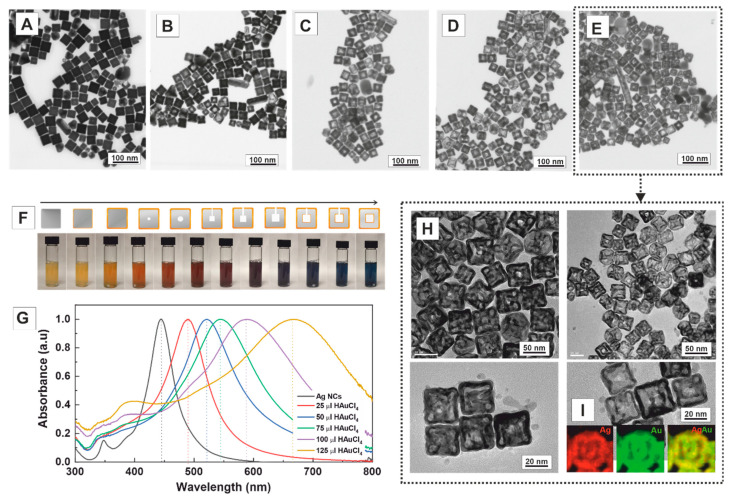
Effect of CTAB on the structural evolution of the Ag nanocubes to multichambered opened AuAg nanoboxes after adding different amounts of HAuCl_4_. Representative transmission electron microscopy (TEM) images of AuAg nanoboxes synthesised in the presence of CTAB with the addition of various amounts of HAuCl_4_ 1 mM: (**A**) 25 μL (6.25 × 10^−4^ mmol), (**B**) 50 μL (1.25 × 10^−3^ mmol), (**C**) 75 μL (1.87 × 10^−3^ mmol), (**D**) 100 μL (2.50 × 10^−3^ mmol), and (**E**) 125 μL (3.12 × 10^−3^ mmol). (**F**) Variation of the colour of the colloidal solution upon increasing amounts of HAuCl_4_ 1 mM added into a fixed volume of Ag NCs in the presence of CTAB 14 mM. (**G**) UV–Vis spectra of Ag nanocubes (black), A (red), B (blue), C (green), D (purple), and E (yellow). (**H**) HR-TEM images of multichambered opened AuAg nanoboxes as shown in (**D**). (**I**) EDS mapping of double-wall nanoboxes (red for Ag, green for Au and composite).

**Figure 3 nanomaterials-13-02590-f003:**
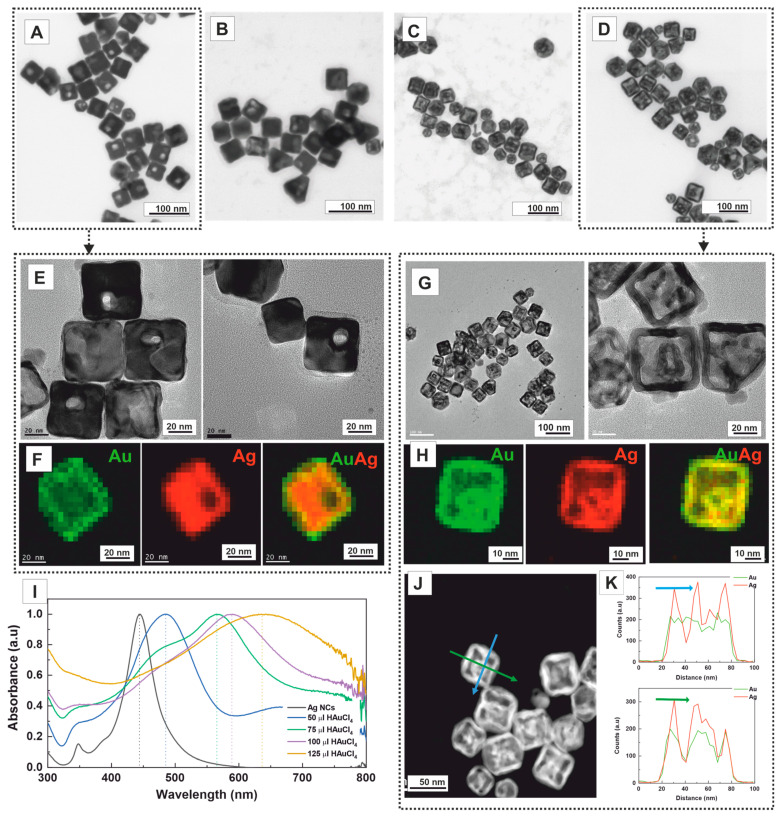
Effect of CTAC on the structural evolution of the Ag nanocubes to multichambered opened AuAg nanoboxes after adding different amounts of HAuCl_4_. Representative transmission electron microscopy (TEM) images of AuAg bimetallic nanoboxes synthesised in the presence of CTAC with the addition of various amounts of HAuCl_4_ 1 mM: (**A**) 50 μL (1.25 × 10^−3^ mmol), (**B**) 75 μL (1.87 × 10^−3^ mmol), (**C**) 100 μL (2.50 × 10^−3^ mmol), and (**D**) 125 μL (3.12 × 10^−3^ mmol). (**E**) HR-TEM images of pinhole AuAg bimetallic nanoboxes as shown in (**A**). (**F**) HAADF-STEM images EDS mapping of pinhole AuAg bimetallic NCs shown in (**A**) (red for Ag, green for Au and composite). (**G**) HR-TEM images of multichambered opened AuAg nanoboxes NCs as shown in (**D**). (**H**) HAADF-STEM images and EDS mapping of multichambered opened AuAg nanoboxes NCs shown in (**D**) (red for Ag, green for Au and composite). (**I**) UV–Vis spectra of Ag nanocubes (black), A (blue), B (green), C (purple), and D (yellow). (**J**) HAADF-STEM image of multichambered opened AuAg nanoboxes shown in (**D**). (**K**) EDS line scanning through the green/blue arrow of nanostructure presented in (**J**).

**Figure 4 nanomaterials-13-02590-f004:**
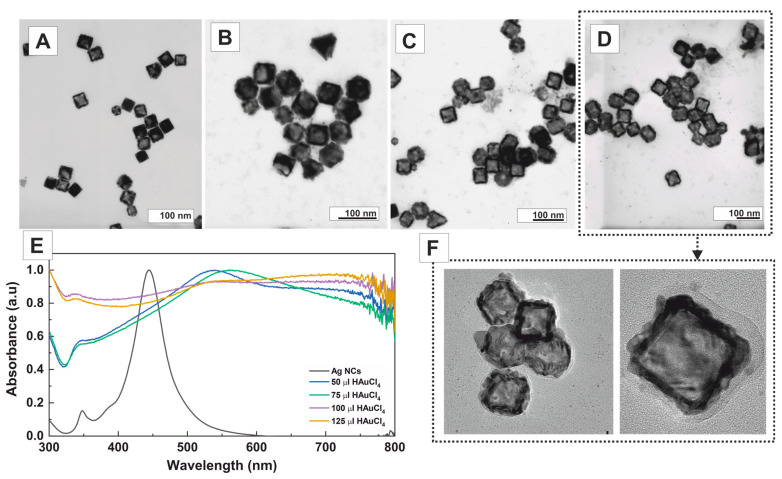
Effect of CTApTS on the structural evolution of the Ag nanocubes to single-walled closed AuAg nanoboxes after adding different amounts of HAuCl_4_: (**A**) 50 μL (1.25 × 10^−3^ mmol), (**B**) 75 μL (1.87 × 10^−3^ mmol), (**C**) 100 μL (2.50 × 10^−3^ mmol), and (**D**) 125 μL (3.12 × 10^−3^ mmol). (**E**) UV–Vis spectra of Ag NCs (black), A (blue), B (green), C (purple), and D (yellow). (**F**) HR-TEM of pinhole AuAg bimetallic nanoboxes as shown in (**D**).

**Figure 5 nanomaterials-13-02590-f005:**
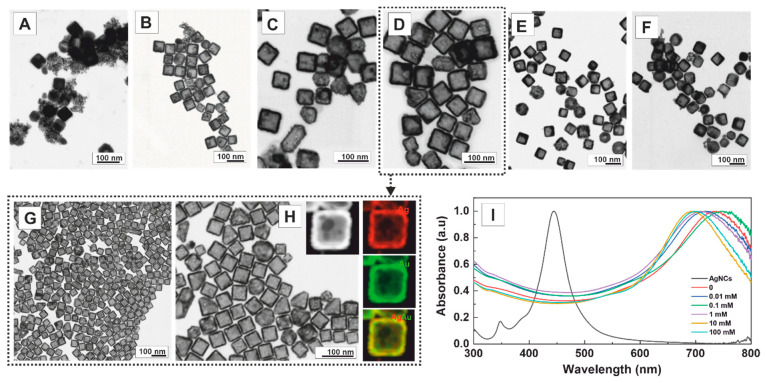
Effect of PVP on the structural evolution of the Ag nanocubes to single-walled closed AuAg nanoboxes. TEM images of AuAg bimetallic nanoboxes synthesised in the presence of increasing concentrations of PVP (by repeating unit) after adding 100 μL of HAuCl_4_ 1 mM (2.50 × 10^−3^ mmol): (**A**) No PVP. (**B**) 0.01 mM, (**C**) 0.1 mM, (**D**) 1 mM, (**E**) 10 mM, and (**F**) 100 mM. (**G**) HAADF-STEM image and (**H**) EDS mapping of a single-walled nanobox shown in (**D**) (red for Ag, green for Au). (**I**) UV–Vis spectra of Ag NCs (black), A (red), B (blue), C (green), D (purple), E (yellow), and F (cyan).

**Figure 6 nanomaterials-13-02590-f006:**
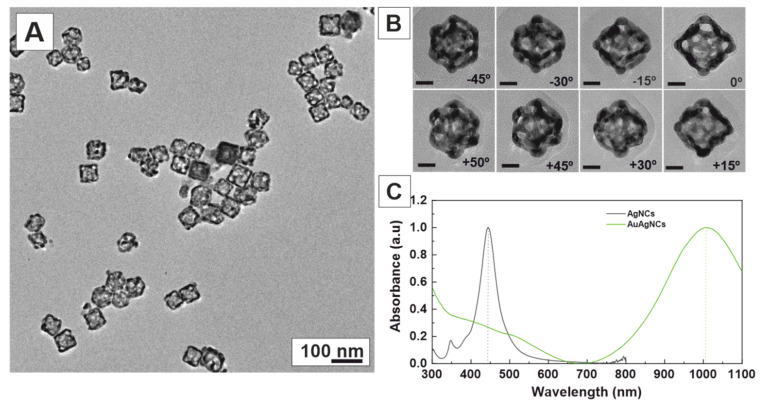
Highly porous AuAg nanoboxes. (**A**) TEM images of AuAg nanoboxes synthesised in the presence of PVP 180 mM (by repeating unit) after the slow addition of 250 μL of HAuCl_4_ 1 mM (10 μL/min). (**B**) TEM images of the same nanobox tilted −45°, −30°, −15°, 0°, 15°, 30°, 45°, and 50°. The scale bar represents 20 nm for all images. (**C**) UV–Vis spectra of Ag NCs (black) and AuAg nanoboxes (green).

## Data Availability

The data that support the findings of this study are available from the corresponding author upon request.

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
