# Peer review of "Sculpting Windows onto AuAg Hollow Cubic Nanocrystals"

_nanomaterials, 2023, doi:10.3390/nano13182590_

Round 1

Reviewer 1 Report

The reviewed paper concerns the optimization of the method of obtaining hollow metal nanocrystals in terms of examining the influence of a number of reagents (surfactants) on the synthesis process and the morphology of the obtained nanoparticles. This is another work of the team on this type of structures. In previous studies, the authors focused on obtaining similar structures and the influence of synthetic conditions on this process, however, this mainly concerned research with other elements such as Pt and Ag, etc. In this respect, the presented results differ in terms of novelty from those previously presented, and have the most in common with the results presented in J. Nanophotonics; 6 (1): 193 - 213. 2017. Although the study of the effects of chloroauric acid appears to be a repetition of earlier results, it is justified due to the greater clarity when comparing the new results. In my opinion, this is a well-planned and well-executed synthetic and physicochemical characterization work. The work certainly makes a significant contribution to the optimization of obtaining analogous hollow structures

Author Response

We thank the reviewer for his/her positive feedback and the recognition of the novelty and significance of our research. It's worth noting that the paper referred to by the reviewer, "J. Nanophotonics; 6 (1): 193 - 213. 2017," is indeed a review paper, primarily centred on general aspects, describing the state of the art on the topic. In this regard, our work differs from that paper because it complemented. We focus on experimental details and discuss the synthetic conditions, providing a more comprehensive understanding of the strategy involved. We emphasize the importance of highlighting this distinction, as it reinforces the originality and contribution of our work.

Reviewer 2 Report

This manuscript reports a systematic study on the effects of surfactants on the opening and pore structure of the Au-Ag bimetallic nanoboxes derived by a Galvanic replacement reaction at room temperature. CTAB and CTAC with Cl or Br counterions are found to contribute to the open of the nanoboxes; whilst the non-halogenated CTApTS led to single-walled closed nanoboxes and the PVP additive favors the formation of the closed nanoboxes with more well-defined structure. The study contributes to the hollow AuAg nanostructures for various applications. The manuscript is well written, with major conclusions supported. I recommend publication after a minor revision.

1. The exact Au-Ag compositions for the samples demonstrated in Figures 2-4 should be specified, which are achieved with different HAuCl4 concentrations. The exact contributions of the various Au/Ag ratio, if exists, to the light response should be discussed.

2. It states that “The EDS compositional analysis (Fig. 5H) revealed that the wall of the nanobox consisted of 37.8 ± 6.2 % Ag and 62.2 ± 6.2 % Au, while the inner part consisted of 36.2 ± 9.2 % Ag and 63.8 ± 9.2 % of Au.”; yet Fig. 5H shows actually EDS mapping.

None

Author Response

  1. The exact Au-Ag compositions for the samples demonstrated in Figures 2-4 should be specified, which are achieved with different HAuCl4 concentrations. The exact contributions of the various Au/Ag ratio, if exists, to the light response should be discussed.

Thank you for your valuable feedback. The composition of the material undergoes a chemical transformation, from pure silver in the initial template to pure gold in the fragmented debris resulting from NP’s corrosion. As the ratio of gold to silver (Au/Ag) increases, the corrosion process follows a well-documented pathway outlined in references 1, 2. Initially, gold is deposited onto the silver nanocrystal template. As the coverage reaches its peak, pinholes begin to emerge, causing the voids to gradually empty until boxes are formed. Subsequent corrosion, either through introducing Au3+ ions or H+, triggers dealloying, resulting in multiple porosities that evolve into nanocages. Finally, full silver oxidation occurs, leading to the collapse of the nanocages and the formation of small Au NPs of 2-5 nanometers in size.

This was already described in the introduction and it has been refined for increased understanding.

“In the process, as time advances or the Au/Ag ratio increases, single or multiple voids open at the NC surface which serve as channels through which the template's core is removed, resulting in the formation of hollow nanostructures. In the initial stages of the process, as the more noble metal is deposited and covers the template, the chemical potential between the deposited and the corroded metals increases. This leads to the formation of a pinhole at the interface, allowing a flux of cations to escape and preventing its closure until the core of the template is voided 1. The number and position of these pinholes determine the final structure of multichambered hollow NCs2. Typically, the pinhole closes once the corrosion of the template is complete, especially when the reaction is conducted at high temperatures.3, 4 Once the hollow NC is formed, for high Au/Ag ratios, further corrosion triggers a de-alloying process that forms pores in the hollow NC surface. As more vacancies collapse, these pores gradually develop, small and polydisperse in size and shape, distributed across the surface. While the shape of the resulting hollow NCs is primary determined by the starting template, wall thickness, composition, and porosity are controlled by the interfacial alloying and de-alloying processes associated with the GRR 5-7.”

  1. It states that “The EDS compositional analysis (Fig. 5H) revealed that the wall of the nanobox consisted of 37.8 ± 6.2 % Ag and 62.2 ± 6.2 % Au, while the inner part consisted of 36.2 ± 9.2 % Ag and 63.8 ± 9.2 % of Au.”; yet Fig. 5H shows actually EDS mapping.

Thank you for bringing this to our attention. The main text has been modified accordingly:

“The EDS elemental mapping shows the distribution of the constituent elements (Fig. 5H) revealing that the wall of the nanobox consisted of 37.8 ± 6.2 % Ag and 62.2 ± 6.2 % Au, while the inner part consisted of 36.2 ± 9.2 % Ag and 63.8 ± 9.2 % of Au.”

Reviewer 3 Report

This manuscript entitled, Sculpting Windows onto AuAg Hollow Cubic Nanocrystals submitted by Javier Patarroyo et al. to the journal Nanomaterials. Authors executed the work in a systematic manner and presented the results with a supporting evidence clearly. Hence my recommendation towards the manuscript is minor revision.

  1. Need to revise English language in some part

This manuscript entitled, Sculpting Windows onto AuAg Hollow Cubic Nanocrystals submitted by Javier Patarroyo et al. to the journal Nanomaterials. Authors executed the work in a systematic manner and presented the results with a supporting evidence clearly. Hence my recommendation towards the manuscript is minor revision.

  1. Need to revise English language in some part

Author Response

We appreciate the reviewer for his/her recommendation and for recognizing the clarity and systematic approach of our work. We have diligently revised the manuscript to enhance the language quality.
